# Identification of Key Endoplasmic Reticulum Stress-Related Genes in Non-Alcoholic Fatty Liver Disease

Zhuang Li, Haozhen Yu and Jun Li *

School of Basic Medical Sciences, Shaanxi University of Chinese Medicine, Xianyang 712046, China
* Correspondence: hello_jack@126.com

**Abstract:** Background: Endoplasmic reticulum stress (ERS) is involved in the etiology of non-alcoholic fatty liver disease (NAFLD). Thus, the current study was designed to identify key ERS-associated genes in NAFLD. Methods: RNA-Seq data of NAFLD and controls were sourced from the Gene Expression Omnibus (GEO) database. Differentially expressed genes (DEGs) in NAFLD and controls were identified by limma. By overlapping DEGs and ERS-related genes, ERS-related DEGs were identified. The function of ERS-related DEGs was characterized by clusterProfiler. Next, the protein–protein interaction (PPI) network was created using the Cytoscape software and the STRING database to identify key ERS-related genes in NAFLD. Furthermore, the correlations among key ERS-related genes were calculated. Results: A total of 8965 DEGs were identified between NAFLD and controls in the GSE126848 dataset. After overlapping these DEGs and ERS-related genes, 20 genes were identified as ERS-related DEGs in NAFLD. Functional analysis revealed that the genes mainly participated in ER-related functions, such as the ER–nucleus signaling pathway, regulation of ERS response, and protein processing in ER. The PPI network revealed the interactions among 17 ERS-related DEGs, including ERN1, ATF6, and EIF2S1 as the key genes. The expressions of ERN1, ATF6, and EIF2S1 were significantly down-regulated in NAFLD and were strongly positively correlated with each other. Further, the expression of ERN1 and ATFA6 was also similar in the GSE89632 datasets. Conclusion: The present study identified ERN1, ATF6, and EIF2S1 as key ERS-related genes in NAFLD. These findings may provide a molecular basis for the role of ERS in NAFLD.

**Keywords:** non-alcoholic fatty liver disease; endoplasmic reticulum stress; protein–protein interaction network; key genes; bioinformatics

## 1. Introduction

With the transformation of human lifestyle and the continuous improvement in life quality in recent years, NAFLD has become the most common cause of hepatic disease. It serves as a driving force in the process of occurrence and development of metabolic diseases such as hypertension and type-2 diabetes. Hence, it has been proposed to rename NAFLD as a metabolic-associated fatty liver disease [1]. The statistics show that its incidence and mortality are increasing each year in developed countries, especially the United States. Due to poor dietary habits being the primary problem in NAFLD, the number of children suffering from this disease is increasing gradually. Around the world, approximately 10–40% of people develop NAFLD; nearly 20% of patients are diagnosed with it in China alone [2,3]. Excluding factors such as diet and lifestyle, the onset of NAFLD is found to be related to genetic factors. These genes can affect fat metabolism and also promote the development of fibrosis and hepatocellular carcinoma (HCC) [4]. Long-term studies have shown that the overall and liver-specific mortality of NAFLD patients is higher compared to the general population [5]. The increased prevalence of NAFLD may be increasing the number of patients with liver cirrhosis and end-stage liver disease [6,7] and the incidences of hepatocellular carcinoma [8].

NAFLD is the general name for a range of hepatic diseases. It is mainly divided into the following three stages: non-alcoholic hepatic steatosis caused by massive fat deposition in the first stage, which gradually evolves into non-alcoholic steatohepatitis (NASH) in the second stage. It can be reversed in the first two stages, but it becomes irreversible during liver cirrhosis and the HCC stage. The deposition of lipids in the liver is the main reason for its onset, which leads to lipid metabolism disorder and steatosis ("the first blow"). Furthermore, it activates hepatic Kuffer cells to release a large number of inflammatory factors, thus, causing an inflammatory reaction and oxidative stress reaction ("the second blow"). Subsequently, mechanisms such as "multiple blows" and "hepato- intestinal axis" were proposed [9].

As an essential metabolic organ, the liver plays an important role in the synthesis and secretion of very low-density lipoprotein as well as the synthesis and metabolism of cholesterol. It is due to the presence of a large amount of hepatic endoplasmic reticulum (ER). The ER is proven to be essential in regulating lipid metabolism in the liver. When the hepatic homeostasis is affected, the ER regulates and restores the normal metabolic function of the liver through ERS and unfolded protein response (UPR). The latter activates transmembrane protein on the ER surface to promote the folding, secretion, and degradation of misfolded proteins in the ER. In addition, ERS also regulates lipid synthesis and metabolism in hepatocytes. It can effectively regulate de novo fat production, insulin sensitivity, and $Ca^{2+}$ homeostasis, which is associated with the occurrence and development of NAFLD [10,11].

ERS targets many pathways in adipose tissue, liver, and pancreas. It affects the browning and heat production of adipose tissue, which is conducive to liver adipogenesis and impairs the secretion of intracellular glucose-stimulated insulin from pancreatic β-cells. Due to the involvement of ERS in the pathogenesis of obesity, hepatic steatosis, and diabetes, it has become a potential target for the treatment and prevention of metabolic diseases. Studies on NAFLD have confirmed that the long-term over-uptake of saturated fatty acid (SFA) leads to an imbalance between protein load and folding in ER. It leads to the over-activation of UPR, thus, activating the cell death pathway. It is inferred that chemical substances that eliminate ERS may have the potential to treat NAFLD [12].

The above studies showed that ERS is closely related to the occurrence of NAFLD. The specific mechanism is not yet clearly understood. Therefore, the current study aims to find the key ERS-related genes in NAFLD to provide alternatives for the clinical treatment of this disease.

## 2. Materials and Methods

### 2.1. Data Used in the Current Study

Thirty-eight ERS-related genes were sourced from a previous study [13]. RNA sequencing data of liver biopsies from 15 NAFLD and 14 healthy normal-weight controls in the GSE126848 dataset were used to identify key ERS-related genes in NAFLD. Additionally, gene expression data of liver samples from 20 NAFLD and 24 healthy controls in GSE89632 were used as an external dataset to characterize the expression patterns of key ERS-related genes.

### 2.2. Exploration of ERS-Related DEGs

DEGs between 15 NAFLD and 14 controls from the GSE126848 dataset were screened by "limma" package with $|log2FC| \geq 1$ and adjust $p$-value $< 0.05$. Then, the Venn algorithm was adopted to detect ERS-related DEGs in NAFLD by overlapping DEGs obtained from GSE126848 with thirty-eight ERS-related genes. The function of ERS-related DEGs was analyzed by the clusterProfiler R package. The molecular functions (MFs), cellular components (CCs), biological processes (BPs), and KEGG pathways with a $p$-value $< 0.05$ were considered significantly enriched.

## 2.3. Identification of Key ERS-Related Genes in NAFLD

ERS-related DEGs were input into the STRING database to develop the PPI network with the confidence set at 0.4. Then, cytoHubba in Cytoscape was applied to screen the important genes in the network using the maximal clique centrality (MCC) algorithm. The top three nodes were defined as key ER-stress-related genes in NAFLD. Furthermore, the correlations among key genes were calculated, and the expression patterns were extracted and compared between NAFLD and controls in GSE89632 by the Wilcoxon test.

## 2.4. Animals and Diets

All animal experiments using male wild-type (WT) C57BL/6 MICE aged 6–8 weeks were approved by the Animal Ethical Laboratory Committee of the Fourth Military Medical University. The mice were placed in a temperature control chamber (22 $\pm$ 2 °C) for 12 h of light/dark cycle, and were free to obtain food and water. After a 2-week acclimatization, normal diet group ($n$ = 6, normal feeding); high-fat diet (HFD)-induced obesity group ($n$ = 6, fed on a high-fat diet and added 23.1 g L-fructose plus 18.9 g L-glucose in water).

## 2.5. Q-PCR Assays

Take a certain amount of liver tissue, add 1 mL of Trizol reagent (Life Technologies, Carlsbad, CA, USA, 15596018), extract RNA by Trizol method, and finally add 20 μL DEPC (Service Bio, Wuhan, China, g3004-100 mL) dissolved in water. Using primescript™ RT Master Mix (perfect real time) (Takara, Kusatsu City, Japan, rr036a) reverse transcripts RNA into cDNA. Using TB green® Premix Ex Taq™ II (TLI RNaseH plus) (Takara, Kusatsu City, Japan, RR820a) for Q-PCR detection.

The gene sequence designed and detected by Tsingke biotechnology is as follows:
Gadph(Mus)-F: GGTGAAGGTCGGIGTGAACG
Gadph(Mus)-R: CTCGCTCCTGGAAGATGGTG
Ern1(Mus)-F: ACACTGCCTGAGACCTTGTTG
Ern1(Mus)-R: GGAGCCCGTCCTCTTGCTA
Atf6(Mus)-F: TCGCCTTTTAGTCCGGTTCTT
Atf6(Mus)-R: GGCTCCATAGGTCTGACTCC
Eif2s1(Mus)-F: TACAAGAGACCTGGATACGGTG
Eif251(Mus)-R: TGGGGTCAAACGCCTATTGATA

## 2.6. Statistical Analysis

Quantitative data were analyzed by Prism 9.0, and were expressed as the mean $\pm$ SEM. Differences between groups were analyzed for statistical significance with Student's two-tailed, unpaired t test, and statistical significance was considered at $p < 0.0001$.

## 3. Results

### 3.1. ERS-Related DEGs Were Identified in NAFLD

A total of 8965 DEGs were detected between NAFLD and control samples in GSE126848 dataset (Table S1). Among these, 42 genes were expressed at higher rates, while 8923 were expressed at lower rates in NAFLD samples relative to controls (Figure 1A). After overlapping with ERS-related genes, 20 ERS-related DEGs were identified in NAFLD (Figure 1B), including BCL2L11 (BCL2-like 11), BFAR (bifunctional apoptosis regulator), DAB2IP (DAB2 interacting protein), ERN1 (endoplasmic reticulum to nucleus signaling 1), UFL1 (UFM1-specific ligase 1), PARP16 (poly(ADP-Ribose) polymerase family member 16), PTPN1 (protein tyrosine phosphatase non-receptor type 1), VAPB (VAMP-associated protein B and C), ATF6 (activating transcription factor 6), BOK (BCL2 family apoptosis regulator BOK), ATF6B (activating transcription factor 6 beta), MBTPS1 (membrane bound transcription factor peptidase, site 1), WFS1 (Wolframin ER transmembrane glycoprotein), MBTPS2 (membrane bound transcription factor peptidase, site 2), EIF2S1 (eukaryotic translation initiation factor 2 subunit alpha), NCK1 (NCK adaptor protein 1), NFE2L2 (NFE2-like BZIP transcription factor 2), NCK2 (NCK adaptor protein 2), PPP1R15B (protein phosphatase 1

regulatory subunit 15B) and QRICH1 (glutamine-rich 1). The expression of all these genes was markedly reduced in NAFLD (Figure 1C,D). Above all, we detected 20 ERS-273-related DEGs which were significantly differentially expressed in normal and NAFLD samples.

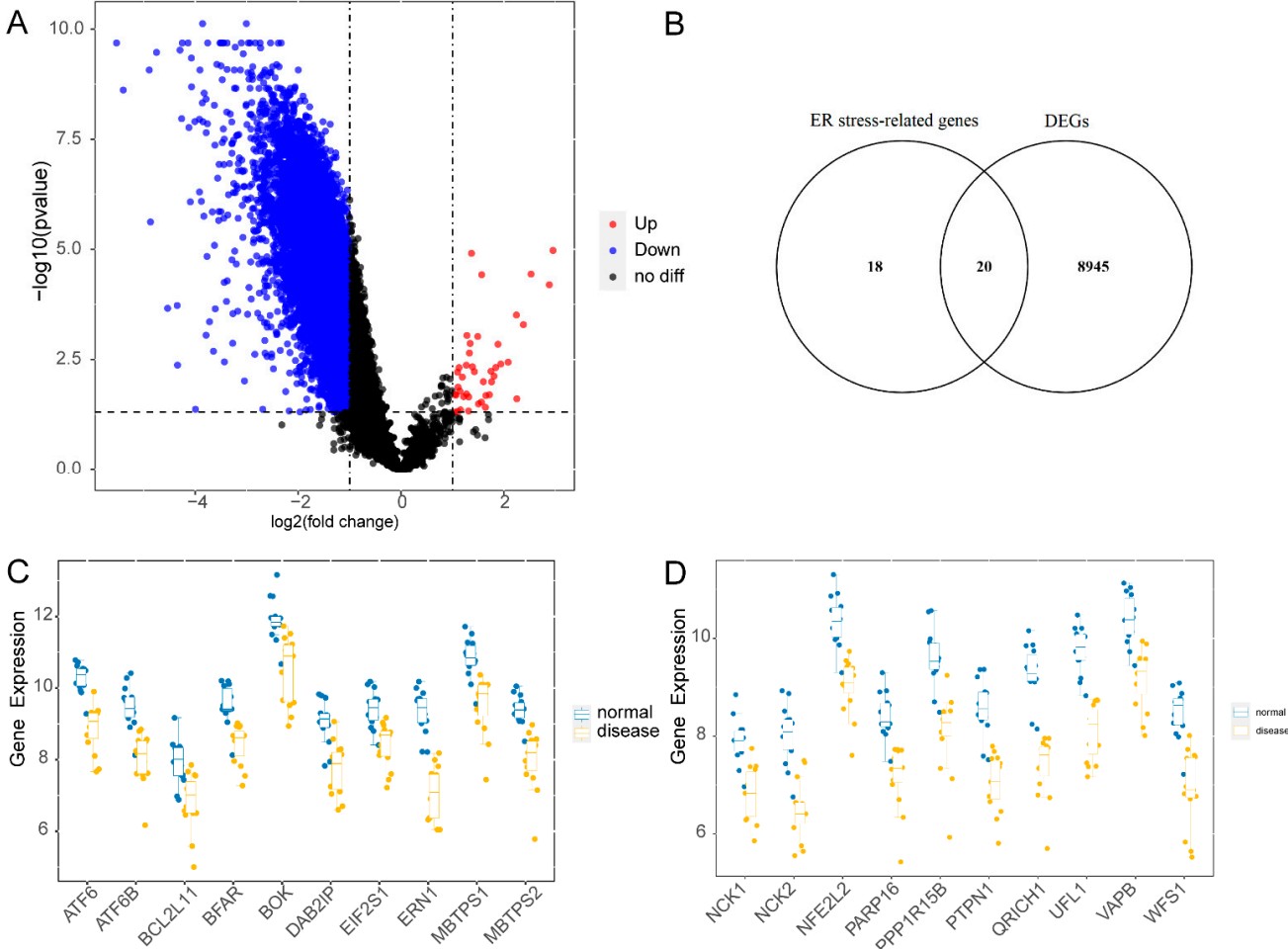

**Figure 1.** (**A**): Volcano diagram of differential gene expression in gse126848. Red and blue dots indicate significant differential expression, red dots indicate up-regulated gene expression in disease samples, blue dots indicate down-regulated gene expression, and black dots indicate no significant difference. (**B**): Veen diagram of the intersection of gse126848 and ER stress-related genes. (**C,D**): Box plot of intersection gene expression difference. The horizontal axis represents the gene, the vertical axis represents the expression, the blue represents the normal sample, and the yellow represents the disease sample.

### 3.2. Functional Analysis of ERS-Related DEGs

The ERS-related DEGs were significantly enriched into ER-related 165 BPs, 8 CCs and 15 MFs (Table S2) including BPs of unfolded protein response, cellular response to unfolded protein, the ER–nucleus signaling pathway, cellular response to topologically incorrect protein, response to unfolded protein, regulation of response to ERS, response to topologically incorrect protein, regulation of ER unfolded protein response, response to ERS, negative regulation of ER unfolded protein response (Figure 2A); CCs of integral component of ER membrane, integral component of organelle membrane, nuclear envelope, intrinsic component of ER membrane, phosphatase complex, intrinsic component of organelle membrane, protein serine/threonine phosphatase complex, protein phosphatase type 1 complex (Figure 2B); and MFs of signaling adaptor activity, cAMP response element-binding, cytoskeletal anchor activity, receptor tyrosine kinase binding, protein–macromolecule adaptor activity, protein phosphatase 2A binding, molecular adap-

tor activity, ephrin receptor binding, cadherin binding, and signaling receptor complex adaptor activity (Figure 2C). Further, these ERS-related DEGs also participate in the KEGG pathways of Parkinson's disease, apoptosis, lipid and atherosclerosis, apoptosis-multiple species, amyotrophic lateral sclerosis, protein processing in ER, and non-alcoholic fatty liver disease (Table S3 and Figure 2D).

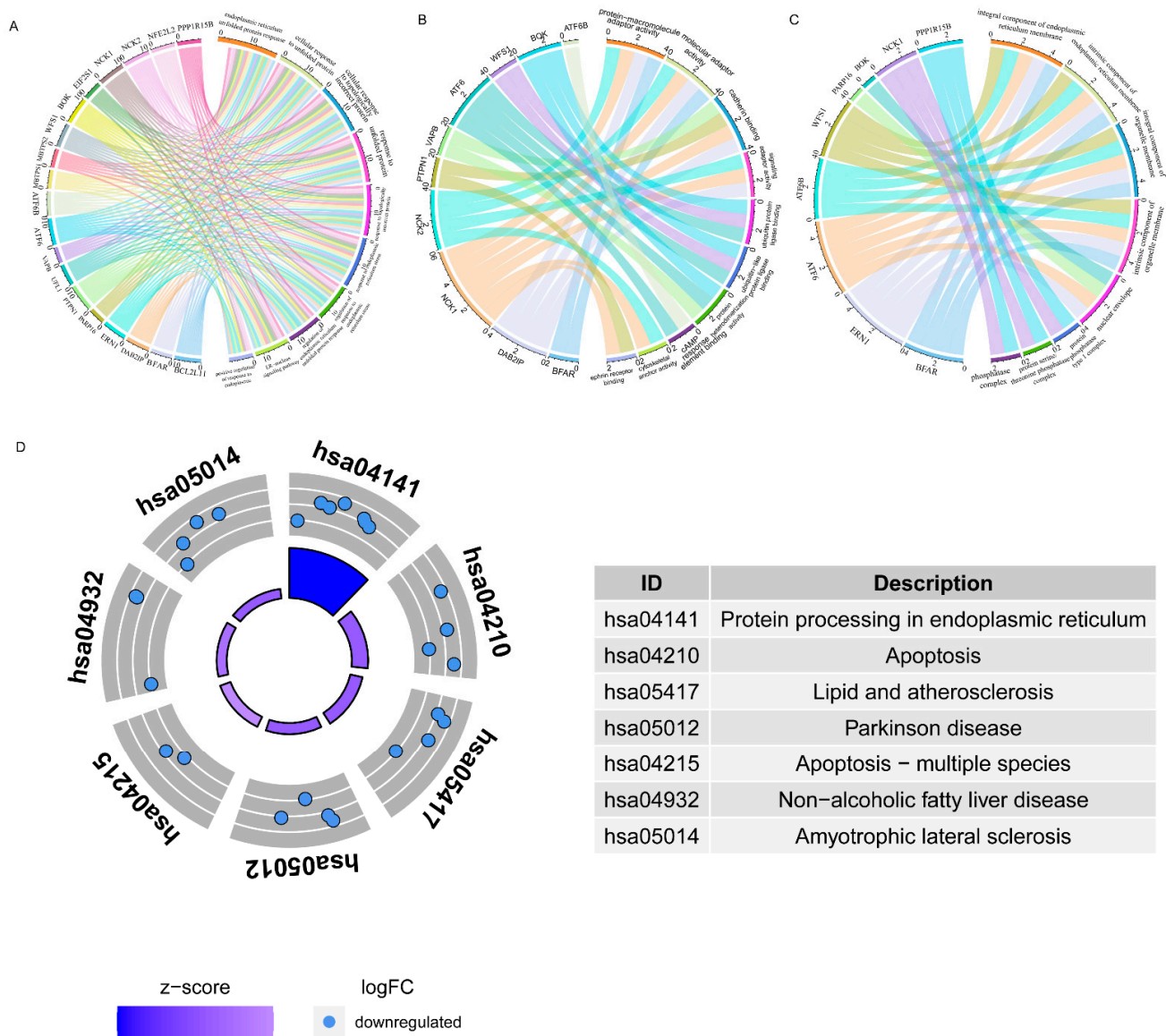

| ID | Description |
|----------|------------------------------------------------|
| hsa04141 | Protein processing in endoplasmic reticulum |
| hsa04210 | Apoptosis |
| hsa05417 | Lipid and atherosclerosis |
| hsa05012 | Parkinson disease |
| hsa04215 | Apoptosis − multiple species |
| hsa04932 | Non−alcoholic fatty liver disease |
| hsa05014 | Amyotrophic lateral sclerosis |

**Figure 2.** (**A–C**): go:bp, go:cc, and go:mf enrichment analysis. (**D**): KEGG enrichment analysis.

### 3.3. ERN1, ATF6, and EIF2S1 Were Key ERS-Related Genes in NAFLD

Seventeen out of twenty key ERS-related genes showed interactions via the STRING database, including BCL2L11, DAB2IP, ERN1, PARP16, PTPN1, VAPB, ATF6, ATF6B, MBTPS1, MBTPS2, WFS1, BOK, EIF2S1, NCK1, NCK2, NFE2L2, and PPP1R15B. Thus, the PPI network by the Cytoscape software was constructed (Figure 3A), in which ERN1, ATF6, and EIF2S1 had the highest scores calculated by the MCC algorithm (Table 1). As a consequence, ERN1, ATF6, and EIF2S1 were identified as key ERS-related genes in NAFLD. The expression patterns of ERN1, ATF6, and EIF2S1 genes showed strong correlations with each other, and the correlation coefficients between ERN1 and ATF6, ATF6 and EIF2S1, and ERN1 and EIF2S1 were calculated to be 0.92, 0.93 and 0.86, respectively (Figure 3B). Consistent expression patterns of ERN1, ATF6, and EIF2S1 were also found in an external

GSE89632 dataset (Figure 3C–E), further demonstrating the important roles of ERN1, ATF6, and EIF2S1 in NAFLD.

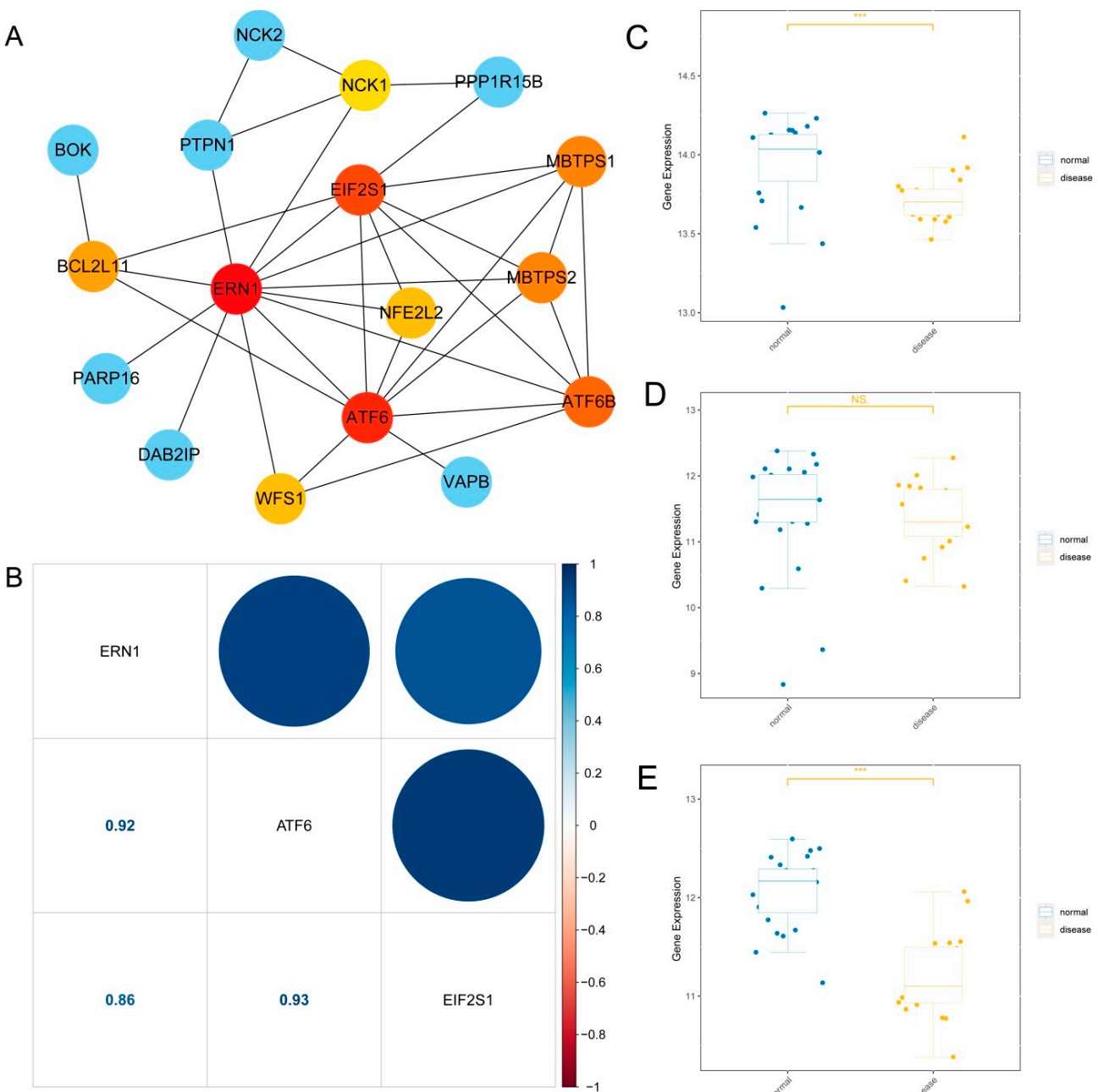

**Figure 3.** (**A**): PPI network of key genes. (**B**): Heat map of key gene correlation. (**C–E**): Box diagram of differential expression of key genes.

**Table 1.** PPI network analysis table.

| Top 17 in Network String-Interactions File Ranked by MCC Method | | |
|:---:|:---:|:---:|
| Rank | Name | Score |
| 1 | ERN1 | 142 |
| 2 | ATF6 | 139 |
| 3 | EIF2S1 | 133 |
| 4 | ATF6B | 126 |
| 5 | MBTPS2 | 120 |
| 5 | MBTPS1 | 120 |
| 7 | BCL2L11 | 7 |
| 8 | NFE2L2 | 6 |
| 8 | WFS1 | 6 |
| 10 | NCK1 | 5 |
| 11 | PTPN1 | 4 |
| 12 | PPP1R15B | 2 |
| 12 | NCK2 | 2 |
| 14 | VAPB | 1 |
| 14 | BOK | 1 |
| 14 | PARP16 | 1 |
| 14 | DAB2IP | 1 |

*3.4. Validation of the Expression of three Key ERS-Related Genes by RT-qPCR*

We performed RT-qPCR for the three genes of interest using livers from mice with non-alcoholic fatty liver disease and found that all three genes of interest showed varying degrees of reduction in expression, with EIF2S1 being the most significantly altered.

**4. Discussion**

Given the yearly increase in incidence and mortality of NAFLD, and the impact of ER on lipid metabolism and insulin sensitivity, the vitality of ER in the occurrence and development of this disease has been proved [11]. In the current study, we screened the key target genes of ER in the pathogenesis of NAFLD, and discovered 20 ERS-related genes. Three key target genes were identified by further analysis. NAFLD mouse model was used for verification, and the same results were obtained as our previous prediction (Figure 4A–C). We enriched NAFLD occurrence and development-related pathways through GO and KEGG, including fat accumulation, lipotoxicity, oxidative stress, mitochondrial dysfunction, and hepato–intestinal axis pathways. After the onset of the disease, the occurrence of insulin resistance leads to an abundant accumulation of free fatty acids in the body. It causes lipotoxicity, cell damage, and inflammatory reaction, thereby changing the normal function of the cells. Because the ER participates in protein synthesis, the stress response occurs after injury of normal cells, followed by induction of autophagy and apoptosis through the c-Jun N-terminal kinase (JNK)-related pathway [14].

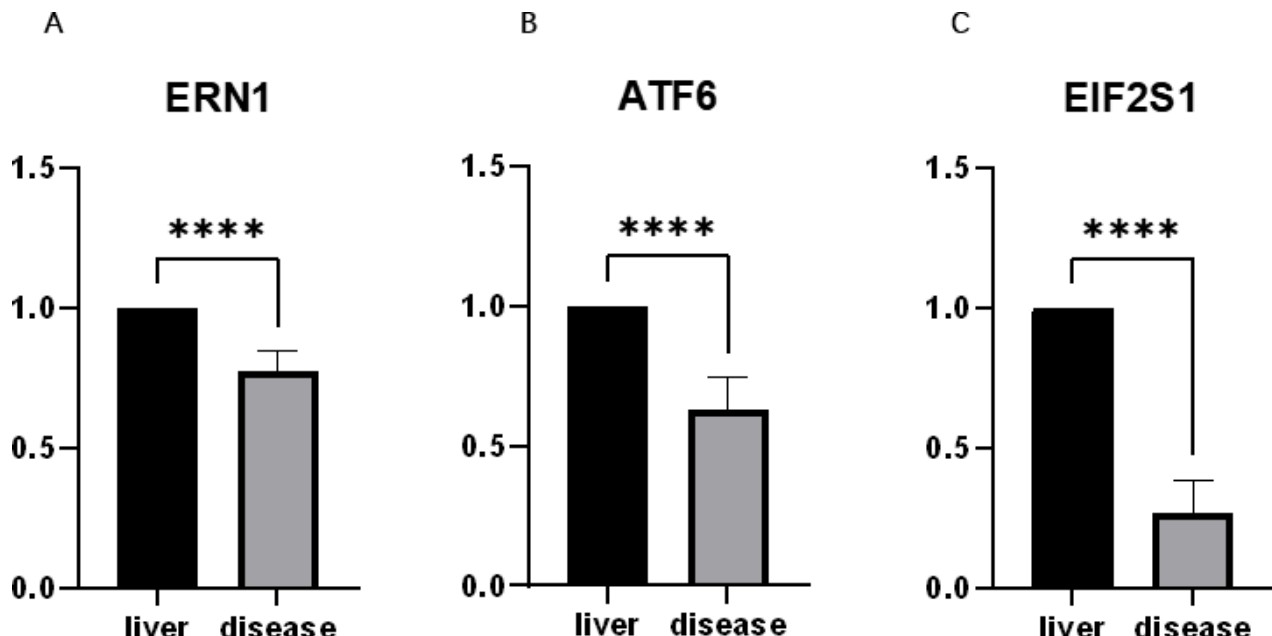

**Figure 4.** (**A**–**C**): PCR verifies the expression difference of three key genes in NAFLD mice. (Data are represented as the mean ± SEM of six independent experiments. **** $p < 0.001$ versus control, versus model by one-way ANOVA with Tukey's test.)

The interactions among the above-mentioned 20 genes were constructed in the protein–protein interaction network, and three key genes were identified (ERN1, ATF6, and EIF2S1). These three key genes are essential for the activation of UPR. In response to ERS, endoplasmic reticulum to nucleus signaling 1 [ERN1, also known as inositol-requiring enzyme 1 (IRE1)] and activating transcription factor 6 (ATF6) are activated. Autophosphorylation and dimerization of ERN1 occur to promote the folding, secretion, and degradation of protein in ER. As the stress response element of ER, ATF6 can promote the transcription of ER-resident proteins, which can inhibit each other via stress-related targets. In addition, ATF6 is also involved in the synthesis of lipids in hepatocytes, which are directly related to the development and prognosis of NAFLD [15–18]. In the UPR, lipid metabolism and steatosis may be regulated through the PERK–eIF2–ATF4 signaling pathway. EIF2$\alpha$ phosphorylation level is particularly important because its complete dephosphorylation leads to steatosis of mouse liver [19,20].

The ER, being an important target in the whole process of occurrence and development of NAFLD, is essential in the lipid metabolism along with the inflammatory response. The involvement of ERS in NAFLD has gradually attracted researchers' attention. Although the role of ER in alleviating hepatic injury through ERS and UPR has been found, it affects inflammatory response and induces autophagy in pathogenesis. The strategies involving ER need to be further explored in the treatment of NAFLD.

With the aid of bioinformatics tools and the use of qPCR to examine three key genes related to ER stress in the livers of mice with NAFLD, we found that three of the key genes screened were significantly differentially expressed in NAFLD mice, and although it is known that there is a strong relationship between ER stress and NAFLD, the specific mechanism of action remains to be thoroughly investigated and explored. It is expected that this screening validation will provide some help for later studies.

**Supplementary Materials:** The following supporting information can be downloaded at: https://www.mdpi.com/article/10.3390/biomedinformatics2030027/s1.

**Author Contributions:** Z.L., H.Y. and J.L. drafted, revised and analyzed the content of this article. All authors contributed to manuscript revision. All authors have read and agreed to the published version of the manuscript.

**Funding:** This research received no external funding.

**Institutional Review Board Statement:** Animal Ethics Committee have a meeting as schedule, the participators have evaluated the care and use of animals described in the protocol of Identification of Key Endoplasmic Reticulum Stress-Related Genes in Non-Alcoholic Fatty Liver Disease and find the procedures described as appropriate and acceptable. Approval No. 20220814001.

**Informed Consent Statement:** Not Applicable.

**Data Availability Statement:** The original contributions presented in this study are included in this article and supplementary materials. Further inquiries can be directed to the corresponding author.

**Conflicts of Interest:** The authors declare that the research was conducted in the absence of any commercial or financial relationships that could be construed as a potential conflict of interest.

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
