# Peer review of "Identification of Key Endoplasmic Reticulum Stress-Related Genes in Non-Alcoholic Fatty Liver Disease"

_biomedinformatics, doi:10.3390/biomedinformatics2030027_

Round 1
Reviewer 1 Report
In this manuscript, the Li et. al. introduced their study of identifying key ERS related genes in non-alcoholic fatty liver disease. The paper is well written and the results are solid. However, there are some questions remain to be answered.
1. Table 1 “string_interactions_short.tsv” need to be substitute with a better name.
2. Figures’ quality are low. Please insert higher resolution ones.
3. Please deposit code (if there is any) to a repository (eg. Github).
Reviewer 2 Report
The article entitled “Identification of Key Endoplasmic Reticulum Stress-Related Genes in Non-Alcoholic Fatty Liver Disease” by Li et al describes the role of endoplasmic reticulum-based stress related genes in Non-Alcoholic Fatty Liver Disease (NAFLD). The reported key genes were identified by performing differential gene expression (DGE) of gene expression data (RNA-seq of NAFLD versus Healthy control; sample size: 15 and 24 respectively) obtained from GSE126848 using Limma. Followed by examining genes specific to ER (list obtained from Liu et al., 2021) and finally validated those significantly expressed genes by comparing with DGE’s from another study (microarray-based gene expression) which was based on non-alcoholic steatohepatitis (NASH) and not NAFLD, (unlike different from what is mentioned in Methods, data obtained from GSE89632). Furthermore, considering DGEs over 1-log2 foldchange brings considerable limitations to the reported genes. The following comments further describes the limitation of this work. The overall work lacks novelty and I don’t feel that this manuscript is suitable for publication in this journal.
Major
- Line 86, 87: Some missing workflows in the Methods limits the understanding of how the authors derived at 8,965 DGEs. Is it that the raw FASTQ files were processed to derive the data, or was the data taken from the available expression matrix (GSE126848), in either scenario how was the transcripts normalized? I.e., was it normalized at RPKM or CPM or any other method? This followed by |log2(fold change)| of ≥ 1 derives at genes that rarely show any biological effect.
- line 137: Follow-up on normalization techniques used and considering the of fold change cut-off, all of overlapping genes identified (n=20) in this study are downregulated which backs the theory of ER stress related – genes. However, except ERN1 majority of the genes show marginal fold differences (14 genes, log2(FC) < 1.5), including key genes namely ATF6 and EIF2S1 discussed in this manuscript to have any biological effect. The same drawback applies to section 3.3 findings.
- All figures (except Figure 4) are highly illegible. Figure 4 should be the discussed in Results.
- The Figure 1C and Figure 1D, y-axis says expression(log2) which has to be properly defined, are these raw read counts or normalized counts (RPKM/FPKM/RPM etc.). In my understanding the authors have compared different disease set (GSE89632), which is not the same disease, not the same sequencing/expression profile platform and lack normalization techniques in comparing both sets and the y-axis description derived from microarray data needs to be mentioned, in Figure 3C-E.
Look at Experiment Type for both datasets used: https://www.ncbi.nlm.nih.gov/geo/query/acc.cgi?acc=GSE126848
https://www.ncbi.nlm.nih.gov/geo/query/acc.cgi - Contradictory statements:
Line 79-82 (Introduction): “The above studies showed that ERS is closely related to the occurrence of NAFLD. The specific mechanism is not yet clearly understood. Therefore, the current study aims to find the key ERS-related genes in NAFLD to provide alternatives for the clinical treatment of this disease.” and
Line 237-242 (Discussion): “Although the three key ERS-related genes in NAFLD identified in this study had been verified in terms of the stress reactions, their expression and interaction between the specific mechanism and other pathways involved in pathogenesis need to be further studied and explored. These three key genes are believed to be important in the occurrence and development of NAFLD. These genes are expected to play a key role in the research and treatment of this disease in the future.
The q-PCR experiment provide differences in the gene expression and the rest is all bioinformatics work. No concluding experiments prove the mechanism in this study which is already defined in literature. For example, the gene ERN1 also known as IRE1α (endoplasmic reticulum to nucleus signaling 1) and ATF6 are reported as role players in NAFLD (Lebeaupin et al).
Lebeaupin et al. (2018). Endoplasmic reticulum stress signalling and the pathogenesis of non-alcoholic fatty liver disease. Journal of Hepatology. 69, 04, 927-947. https://doi.org/10.1016/j.jhep.2018.06.008.
6. All the R-scripts/algorithms used should be made available as supplementary materials.
Minor
- Line 33, p-value < 0.05 should be corrected to represent adjusted p-value
- Line 135, 136, higher and lower rates must be defined
- All mention of gene names must be italicized and represented in upper case for humans and Camel case for mouse.
- Line 107, approval number should be mentioned “Animal Ethical Laboratory Committee of the Fourth Military Medical University”
- Missing corresponding authors affiliation. All authors affiliation should be mentioned appropriately.

Round 2
Reviewer 2 Report
The article entitled “Identification of Key Endoplasmic Reticulum Stress-Related Genes in Non-Alcoholic Fatty Liver Disease” by Li et al describes the role of endoplasmic reticulum-based stress related genes in Non-Alcoholic Fatty Liver Disease (NAFLD). The reported key genes were identified by performing differential gene expression (DGE) of gene expression data (RNA-seq of NAFLD versus Healthy control; sample size: 15 and 24 respectively) obtained from GSE126848 using Limma. Followed by examining genes specific to ER (list obtained from Liu et al., 2021) and finally validated those significantly expressed genes comparing with another dataset and in-vitro. The authors have revised the manuscript considerably:
Major:
1. It is confusing, as to why the data with nonalcoholic steatohepatitis (NASH) was chosen from data (GSE89632) and represented to be 19 NAFLD to support the results, while the actual work is purely based on 19 nonalcoholic steatohepatitis (NASH) and not simple steatosis (NAFL). If this was the goal, then why was NASH data (n=16) omitted from the original dataset (GSE126848) during the analysis. As such the context is based on NAFL and not NASH. The authors are requested to provide a suitable rationale.
Minor:
1. Since they all belong to same department and Institution, mentioning their affiliation once would be sufficient. For example:
Zhuang Li1, Haozhen Yu1 and Jun Li1*
1School of BasicMedical Sciences, Shaanxi University of Chinese Medicine, Xianyang 712046, Shaanxi, China
* Correspondence Email: [email protected] (request to provide institutional email address if any).
2. Line 32: Since the starting dataset was GSE126848, the authors should rewrite the sentence from “Besides, their expression patterns were also similar in the GSE126848 and GSE89632 datasets” to “Besides, their expression patterns were also similar in the dataset GSE89632”.
3. All the resources for bioinformatics analysis should be cited throughout the manuscript, example usage of STRING database, Cytoscape, KEGG etc.
- Line 106, approval number should be mentioned “Animal Ethical Laboratory Committee of the Fourth Military Medical University”
5. Line 164, the genes must be in italics and Camel case for mouse.
6. Figure 3. C-E, y-axis should be appropriately labelled. (Not expression(log2))
Author Response
Please see the attachment.

This manuscript is a resubmission of an earlier submission. The following is a list of the peer review reports and author responses from that submission.